# Telehome Monitoring of Symptoms and Lung Function in Children with Asthma

**DOI:** 10.3390/healthcare10061131

**Published:** 2022-06-17

**Authors:** Audrey Fossati, Caroline Challier, Aman Allah Dalhoumi, Javier Rose, Annick Robinson, Caroline Perisson, François Galode, Baptiste Luaces, Michael Fayon

**Affiliations:** 1Centre Hospitalier Universitaire (CHU) de Bordeaux, Pediatrics Department, Pediatric Pulmonology, CEDEX, 33076 Bordeaux, France; challier.caroline@hotmail.fr (C.C.); francois.galode@chu-bordeaux.fr (F.G.); michael.fayon@chu-bordeaux.fr (M.F.); 2Centre Hospitalier d’Agen-Nérac, Pediatrics Department, 47000 Agen, France; dalhoumia@ch-agen-nerac.fr (A.A.D.); docteur-luaces@hotmail.fr (B.L.); 3Paediatric Department, Seychelles Hospital, Victoria P.O. Box 52, Seychelles; xvrrose@yahoo.co.uk; 4Centre Hospitalier Universitaire Mère Enfant Tsaralàlana, Department of Child Health, Teaching Hospital, Antananarivo 3GVF+76F, Madagascar; annicklalaina@yahoo.fr; 5Centre Hospitalier Universitaire Réunion Sud, Service de Pédiatrie, 97410 Saint Pierre, France; caroline.perisson@chu-reunion.fr; 6Centre de Recherche Cardio-Thoracique de Bordeaux, INSERM U1045, Université de Bordeaux, 33000 Bordeaux, France; 7Centre d’Investigation Clinique (CIC1401), INSERM, 33076 Bordeaux, France

**Keywords:** asthma, child, telemonitoring, spirometry, symptom perception, profile

## Abstract

Background: The ability to perceive bronchial obstruction is variable in asthma. This is one of the main causes of inaccurate asthma control assessment, on which therapeutic strategies are based. Objective: Primary: To evaluate the ability of physicians to characterize the bronchial obstruction perception profile in asthmatic children using a clinical and spiro-metric telemonitoring device. Secondary: To evaluate its impact on asthma management (control, treatment, respiratory function variability) and the acceptability of this telemonitoring system. Methods: 26 asthmatic children aged 6–18 years equipped with a portable spirometer and a smartphone application were home-monitored remotely for 3 months. Clinical and spiro-metric data were automatically transmitted to a secure internet platform. By analyzing these data, three physicians blindly and independently classified the patients according to their perception profile. The impact of telemonitoring on the quantitative data was assessed at the beginning (T0) and end (T3 months) of telemonitoring, using matched statistical tests. Results: Patients could initially be classified according to their perception profile, with a concordance between the three observers of 64% (kappa coefficient: 0.55, 95%CI [0.39; 0.71]). After discussion among the observers, consensus was reached for all patients but one. There was a significant >40% decrease in FEV1 and PEF variability, with good acceptance of the device. Conclusions: Clinical and spiro-metric tele-home monitoring is applicable and can help define the perception profile of bronchial obstruction in asthmatic children. The device was generally well accepted.

## 1. Introduction

Telemedicine is defined by The World Health Organization (WHO, Geneva, Switzerland) as “The delivery of health care services, where distance is a critical factor, by all health care professionals using information and communication technologies for the exchange of valid information for diagnosis, treatment and prevention of disease and injuries, research and evaluation, and for the continuing education of health care providers” (WHO, 2009). It has been a rapidly growing field over the last twenty years, particularly since the COVID-19 epidemic [1,2]. Its importance has been recognized both nationally and internationally [3,4,5,6]. In developing countries, it can be effective in reducing inequities in access to care by enabling remote care delivery, particularly in areas underserved by health facilities [7]. However, the implementation of these technologies is often hampered by various ethical and legal issues [8,9].

Telemonitoring in particular is defined as “an automated process for the transmission of data on a patient’s health status from home to the respective health care setting” [10]. It is increasingly used to monitor chronic patients, and allows the recording of clinical or technical indicators at home with the identification of alerts [11].

The ability to perceive the onset and severity of symptoms of asthma varies among children and depends on multifactorial causes and very complex and largely undefined psycho-physiological mechanisms [12,13]. The use of peak flow meters (calibrated instruments used to measure lung capacity in monitoring breathing disorders such as asthma) is recommended to help the patient objectify the degree of bronchial obstruction, but previous studies have shown that there is only a weak correlation between objective measures of respiratory distress and the subjective dyspnea sensation described by the patient [14,15,16]. Some patients will report significant discomfort with minimal bronchoconstriction, leading the clinician to question whether these symptoms are due to bronchospasm, anxiety or other factors [17], and others will not report symptoms even in the presence of severe obstruction [18].

With this in mind, Brouwer et al. conducted a study in 36 children with mild to moderate persistent asthma who electronically recorded Peak Expiratory Flow (PEF, the maximum flow rate generated during a forceful exhalation, starting from full inspiration) and Forced Expiratory Volume in one second (FEV1, the volume of air (in liters) exhaled in the first second during forced exhalation after maximal inspiration) twice daily for 3 months using a home spirometer [19]. The results showed a poor correlation between the spiro-metric data and clinical disease activity scores. Importantly, the authors were able to distinguish four perception profiles in the asthmatic child: poor perceivers (no symptoms in the presence of severe obstruction); good perceivers (correlation between symptoms and bronchial obstruction); excessive perceivers (many symptoms with no or minimal obstruction) and anarchic perceivers (no correlation between symptoms and obstruction). Since symptom reporting is an integral part of the therapeutic management of asthma (control scores), this may have implications for clinical and therapeutic approaches for these patients. Under-perception may delay the diagnosis and treatment of exacerbations, resulting in a greater risk of morbidity and mortality [20,21]. Conversely, excessive perception of symptoms may lead to overuse of medication and frequent recourse to consultations [22]. The lack of precise determination of these profiles could explain why some studies have not demonstrated a net benefit related to the application of telemonitoring in childhood asthma, particularly in terms of reduction of medical treatments [23].

Therefore, the current study was conducted to assess whether the perception profile of bronchial obstruction of children with asthma can be routinely identified using home telemonitoring of clinical symptoms and FEV1. In addition, we evaluated the impact of home telemonitoring on asthma management (control, therapeutic optimization, spiro-metric signal variability), as well as the acceptability and barriers to the use of this mobile health care system.

## 2. Materials and Methods

### 2.1. Context and Ethics

This was a retrospective study that analyzed data from asthmatic children and adolescents at the Children’s Hospital of the Centre Hospitalier Universitaire (CHU) of Bordeaux and the Seychelles Hospital between December 2018 and January 2021. The children and their parents were informed of the objectives of the study and written informed consent for the use of their data was obtained. In view of the documents at its disposal, the Publication Group of the Ethics Committee of the CHU of Bordeaux issued a favorable opinion on the publication of this research work (Opinion CE-GP-2021/12).

### 2.2. Sample

The study included 26 asthmatic children aged 6 to 11 years old and adolescents aged 11 to 18 years old. The inclusion criteria were: asthma diagnosed by a physician for more than 6 months, moderate to severe persistent asthma according to the Global Initiative for Asthma (GINA) (treatment level ≥ 3), having a home connected tool (tablet and/or smartphone), being able to perform correct spirometry. The exclusion criteria were any other pathology responsible for respiratory symptoms (cystic fibrosis, primary ciliary dyskinesia, chronic obstructive pulmonary disease).

### 2.3. Objectives

The primary objective was to evaluate the physicians’ ability to characterize patients according to Brouwer’s profiles based on data collected by the telemonitoring device. The secondary objectives were to study the impact of home telemonitoring on asthma control, treatment levels (GINA steps), lung function and its variability, and the satisfaction of children and adolescents and parents regarding these telemonitoring devices and process.

### 2.4. Conduct of the Study

As part of routine care, patients received a free electronic spirometer Spirobank Smart^®^ (MIR Company, Langlade, France). This enables the measurement of FEV1, PEF, and FEF25-75 (Forced Expiratory Flow at 25–75% of the vital capacity) which are then transmitted in real time via the free application Pneumotel uploader^®^ (Company LAMIRAU Ingénierie, Langlade, France) installed on the patient’s smartphone, to the central and secure Pneumotel^®^ internet platform. The application also allows the patient to enter his/her clinical symptoms. The clinical symptoms monitored were those related to an exacerbation: wheezing in the chest, shortness of breath, difficulty speaking, more coughing than usual; and those of asthma control according to GINA guidelines: activity limitation, taking bronchodilators (e.g., salbutamol), signs of daytime asthma, nocturnal awakening due to asthma.

The monitoring process began with a 10-day observation phase, during which the patient made twice-daily spiro-metric recordings at home, as well as recording his/her clinical symptoms. In the absence of signs of exacerbation, the best FEV1 value obtained during this period was taken as a reference. Once the observation phase was over, FEV1 and clinical signs were performed at least twice a week outside of exacerbation phases, and daily if the patient thought he had signs of exacerbation. In case of worsening (at least one clinical sign of exacerbation and/or uncontrolled asthma (≥3 items of poor GINA control) and/or a drop in FEV1 ≥ 30% compared to the reference value [24]), the physician received an alert by e-mail and Short Message Service (SMS). They then contacted the patient by phone call or e-mail within 24 h for a real assessment of the situation and to optimize management. The data analyzed included the recordings made during 3 months of follow-up for each patient.

Personal and family history, asthma characteristics, Childhood Asthma Control Test (c-ACT) score and treatments were collected from the computerized hospital medical records. A semi-structured interview was conducted between the physician and the patient and his/her parents after 3 months of use of the device to assess the acceptability and barriers to the use of Spirobank Smart^®^. The After Scenario Questionnaire (ASQ) was also completed to qualitatively assess satisfaction with this connected device [25] (Figure 1).

### 2.5. Statistical Analysis

Data are reported as mean ± standard deviation or median (interquartile range IQR1; IQR3). Statistical analysis was performed using PRISM software (GraphPad Software, San Diego, CA, USA, California 92108). Regarding the identification of the perception profile, 3 physicians (observer 1: M.F.; 35 years of post-thesis experience; observer 2: F.G.; 6 years of post-thesis experience; observer 3: A.F.; in thesis year) classified the patients blindly and independently according to 4 categories: “good perceiver”, “poor perceiver”, “anarchic perceiver”, or “excessive perceiver”. This classification was performed by physicians by comparing on the platform the curves of the reported symptoms and the recorded FEV1 for each patient. The physicians deemed that patients were “good perceivers” if each and every episode of clinical signs of exacerbation were accompanied by significant changes in FEV1 or *vice versa*; “poor perceivers” whenever significant changes in FEV1 were never accompanied by patients’ transmission of clinical signs of loss of control and/or exacerbation; or “excessive perceivers” if patients’ transmission of clinical signs of loss of asthma control and/or exacerbation were never accompanied by significant FEV1 changes. “Anarchic perceivers” were defined as any combination of the 3 profiles in the same patient. The concordance of this classification between the 3 physicians was then evaluated according to the Randolph’s Kappa coefficient method [26]., before and after a collegial discussion of the cases (Delphi method). Agreement by pairs was assessed using Cohen’s Kappa coefficient. The variability of FEV1 and PEF was calculated for each patient over the first 15 and last 15 days of the study using the following formula: (maximum value–minimum value)/mean of the 2 values. The quantitative variables for the two groups were compared by the Student’s *t*-test (or Mann and Whitney for non-Gaussian data). The non-continuous variables were compared by the chi-squared test or two-tailed Fischer exact test (non-parametric). A *p* < 0.05 was considered a statistically significant difference for all tests.

The qualitative study regarding the satisfaction with the telemonitoring device was conducted according to a semio-pragmatic phenomenological interpretative method [27,28]. All the interviews were transcribed word for word, and then analyzed to identify themes. A triangulation of the qualitative data was carried out.

## 3. Results

### 3.1. Study Flow Diagram

The flow chart is shown in Figure 2.

### 3.2. Demographic Characteristics of the Study Population

Demographic characteristics of the study population are shown in Table 1.

### 3.3. Main Objective: Brouwer’s Asthma Profile

After evaluating each patient’s profile independently and blindly, based on the criteria reported by Brouwer et al., the overall concordance between the three observers was 64%. Randolph’s kappa coefficient was 0.55 [0.39; 0.71]. When assessed in pairs, Cohen’s kappa coefficient ranged from 0.33 to 0.61 (Figure 3).

Using a Delphi approach (consensus among observers), the final overall agreement was 97% with a Randolph’s kappa coefficient of 0.97 (0.91; 1.00). The percentage of patients that could be classifiable into a specific profile was 88% (23/26). 38% were anarchic perceivers (10/26), 27% were poor perceivers (7/26), 15% were good perceivers (4/26) and 8% were excessive perceivers (2/26). Two patients were defined as “unclassifiable” because the first patient did not provide enough data, and the quality of the spirometry recordings of the second patient was considered to be poor. There was no consensus for one child (Figure 4; Table 2).

In addition, the data from a further 16 patients were analyzed. These 16 consecutive patients were those in whom the tele-monitoring data and curves were reliably available, but not all the data corresponding to the secondary objectives. In the 42 patients, the overall concordance between the three observers was similar, i.e., 64%. Randolph’s kappa coefficient remained at 0.55 (0.42; 0.69). After the Delphi approach, the final overall agreement was 98% with a Randolph’s kappa coefficient of 0.98 (0.94; 1.00). The percentage of patients who could be classified into a specific profile was 90% (38/42). Forty one per cent of patients were anarchic perceivers (17/42), 24% were good perceivers (10/42), 21% were poor perceivers (9/42) and 5% were excessive perceivers (2/42). One patient was defined as “unclassifiable”.

### 3.4. Secondary Objectives

#### 3.4.1. Asthma Control

There was a non-significant trend towards improvement in the c-ACT score between baseline (median 16 [14; 20]) and the end of the study (median 20 [15; 23]) (Figure 5). The change in the c-ACT scores according to the patients’ perception profile is represented in Figure 6.

#### 3.4.2. Therapeutic Optimization

The distribution of treatment steps from the beginning to the end of the study is shown in Figure 7. The distribution according to the patients’ perception profile is represented in Figure 8.

#### 3.4.3. FEV1 and PEF Variability

The mean FEV1 for the first 15 days (1.49 L/s ± 0.64) did not differ significantly from the mean for the last 15 days (1.48 L/s ± 0.66). The mean PEF for the first 15 days (3.15 L/s ± 1.42) did not differ significantly from the mean for the last 15 days (3.22 L/s ± 1.61).

FEV1 variability decreased from a median of 75.6% (42.6; 87.9) at the beginning of the study to 35.6% (22.7; 73.4) at the end of the study (*p* = 0.006) (Figure 9a). PEF variability decreased from a median of 90.2% (49.6; 112.7) at baseline to 44.4% (19.3; 97.5) at the end of the study (*p* = 0.03) (Figure 9b).

#### 3.4.4. Tool Observance and Acceptability

During the first 10 days, 73% of patients (19/26) recorded at least half of the 20 expected measurements (Figure 10). After these 10 days, 73% of patients (19/26) achieved the expected minimum of 2 weekly recordings.

Children and their parents were generally very satisfied with Spirobank Smart^®^ follow-up according to the ASQ (After Scenario Questionnaire) (Figure 11).

#### 3.4.5. Qualitative Analysis

A total of 15 children and 17 parents were included in the qualitative analysis at the end of the study. When asked, “Were you satisfied with the Spirobank Smart^®^?”, the entire sample was satisfied with the tool and the follow-up. Regarding the positive points reported by children and parents, the system was particularly appreciated for its playful and intuitive aspect. In addition, some children and parents reported a better perception of the severity of asthma exacerbations thanks to the FEV1 and PEF values displayed on the mobile application. Parents expressed a feeling of comfort and reassurance thanks to the telemonitoring. They felt that the monitoring was close without appearing over-medicalized. To the question: “Did you find the device too medicalized?” all but one of the parents answered in the negative. Nevertheless, some parents expressed their anxiety whenever the medical team did not respond quickly enough or if the child left home without his portable spirometer. Half of the parents surveyed did not find the device constraining, but the need to perform the measurement daily during the observation period and during exacerbations was experienced as a constraint by some children. The length of the expired breaths required to record technically good spiro-metric tests also discouraged some children.

## 4. Discussion

The current study showed that clinical and spiro-metric home telemonitoring was applicable. In the majority of cases, it was possible to define the profile of clinical perception of bronchial obstruction in asthmatic children. In the first round, the more experienced investigator recorded more “Anarchic” perceivers; while the younger investigators recorded more “Poor” perceivers. Collegial discussion using the Delphi approach among the professionals resulted in a good level of agreement. In addition, there was a non-significant trend towards improved asthma control (c-ACT score) after 3 months of follow-up, as well as a significant decrease in FEV1 and PEF variability. The observed changes in the distribution of treatment levels were guided by the patients’ perception profile. The device was well accepted by the asthmatic children and their family.

Among the 23 out of 26 children who could be classified, the distribution of perception profiles was as follows: anarchic perceivers (44%) > poor perceivers (30%) > good perceivers (17%) > excessive perceivers (9%). The study of Brouwer et al. of 36 asthmatic children found the following distribution: poor perceivers (36%) > anarchic perceivers (25%) > good perceivers (19.5%) = excessive perceivers (19.5%). These differences could be explained by the severity of the asthma being monitored (less severe in their population) or the way in which symptoms are recorded (written diary versus electronically). The existence of different perception profiles could explain why some studies have not demonstrated a net benefit from the application of telemonitoring of childhood asthma [29], and could have an implication in the design of clinical trials by selecting a certain category of perception profile as the study population. The classification of patients according to their perception profile of bronchial obstruction could also have practical applications by enabling healthcare professionals to propose personalized management to promote optimal disease control. Patients defined as poor perceivers could more easily have their background treatment increased, and conversely a decrease in treatment could be envisaged in over-perceiver patients. This aid to therapeutic adaptation was one of the major expectations of the patients in our study. Finally, it could make it possible to target the profiles that would benefit most from objective measurements of airway obstruction over the long term. Once a patient has been identified as a good perceiver, follow-up could be simplified based on the evaluation of symptoms alone. On the contrary, patients who are poor perceivers could benefit from the use of a peak flow meter or a portable spirometer at home over the long term.

Children and their parents were generally very satisfied with the follow-up. They appreciated the playfulness and ease of use of the device. This enthusiasm for connected devices is part of the era of smart-medicine where more and more devices, gadgets, and applications are being offered to patients [30]. However, the clinical effectiveness of most of these technology-based strategies is not evidence-based and further studies are needed to assess their reliability. Most parents appreciated having access to their children’s spirometry results. This highlights patients’ desire to be actors in their own therapeutic management, a desire that increases with the rate of health crises [31,32]. Nevertheless, they reported a feeling of reassurance from this close medical follow-up with the Spirobank Smart^®^, even to the point of apprehension when it was stopped. Despite the advice given to parents to be careful to maintain their usual lifestyle, the introduction of the device inevitably created a need, even a form of dependency.

Only moderate agreement was found between the authors when attempting to discriminate the patients between the categories established by Brouwer et al. Because there is no standardization of these perception profiles, every physician can see a different pattern using the same data. Therefore, it would be interesting to find a machine learning algorithm that could identify the perception profile independently of the clinician’s evaluation.

The methods used in this study have several limitations. First is the time constraint. Although no performed refused to participate, in the end, three patients refused to fully participate in the Spirobank Smart^®^ follow-up and one patient did not take any measurements at home. The reasons cited were a lack of time and the constraint of using a new device in addition to daily treatment. The implementation of a new therapeutic object in the patient’s daily life should therefore not add too great of a burden [33]. The recommended frequency of use of the device should also be taken into account. Indeed, some children in our study mentioned the constraint of having to perform the measurement daily during the first 10 days. To stimulate compliance over the long term, automatic reminders were sent out after 7 days without a recorded value. A decrease in compliance is found in longer-term studies, such as in the study from Côté et al., where compliance with daily spirometry measurements dropped from 63% in the first month to 33% at 12 months [34]. Second, a decrease in way patients were performing correct spiro-metric measurements during the 3 months of follow-up was observed in some patients in our study. Other studies do not report a decrease in the technical quality of maneuvers over time [35]. Third, this device is not adapted to most children under 6 years old because of prolonged forced expiration. Regular sessions of therapeutic education in the classroom or in e-TPE (Therapeutic Patient Education) are therefore necessary to reinforce compliance and technique. In our cohort, a 4-year-old child is being followed since he can produce satisfactory flow-volume curves. To compensate for age and technical requirements, the usefulness of remote monitoring devices that do not require the active participation of the patient should be studied. We are now in the process of equipping patients with artificial intelligence based electronic stethoscopes (https://www.stethome.com, accessed on 1 May 2022). Lastly, the lack of significant improvement in asthma control could be partly explained by poor adherence to treatment, one of the major causes of uncontrolled asthma [36]. The combination of this system with the use of connected inhalers allowing the collection of adherence data on the same platform could be an even more comprehensive remote monitoring tool [37]. It is clear that all these devices allow for reliable distant monitoring of the lung status of asthmatic children, but how comprehensive (number of devices) they are used will depend on the cost, motivation of patients, and the availability of a dedicated healthcare team.

In conclusion, the current study shows that clinical and spiro-metric home telemonitoring is applicable and can be used to characterize the perception profile of bronchial obstruction in asthmatic children to help adapt therapeutic management. However, this requires an in-depth discussion within the team. Research projects studying the role of such a telemonitoring system on a longer term basis and including other clinical evaluation criteria such as quality of life, exacerbation, unscheduled visits and hospitalization remain to be conducted.

## Figures and Tables

**Figure 1 healthcare-10-01131-f001:**
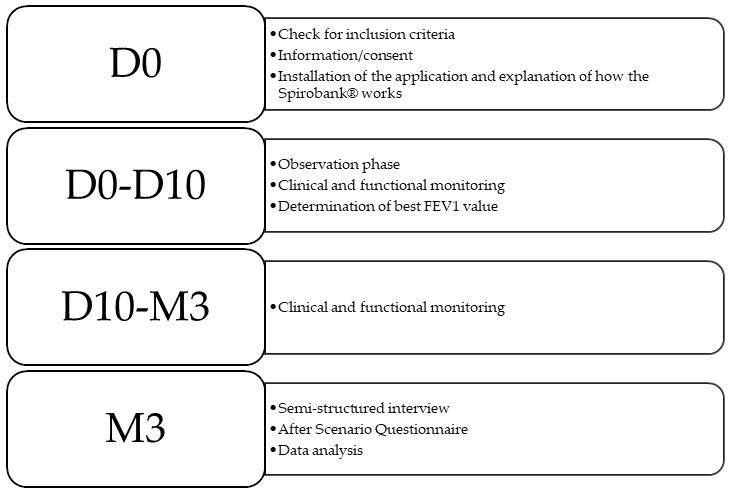
Study methodology.

**Figure 2 healthcare-10-01131-f002:**
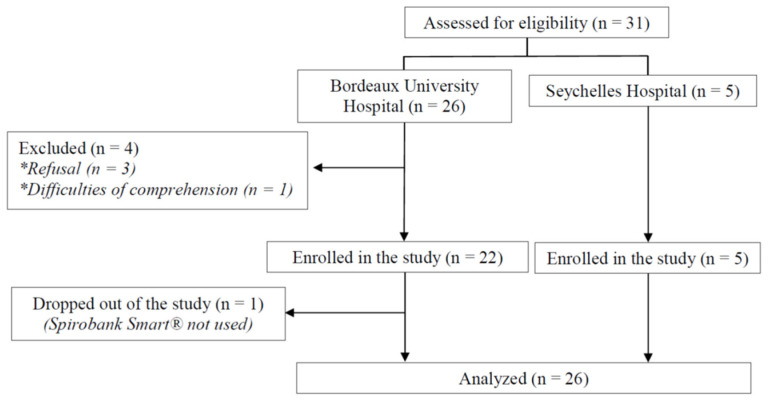
Study flow diagram.

**Figure 3 healthcare-10-01131-f003:**
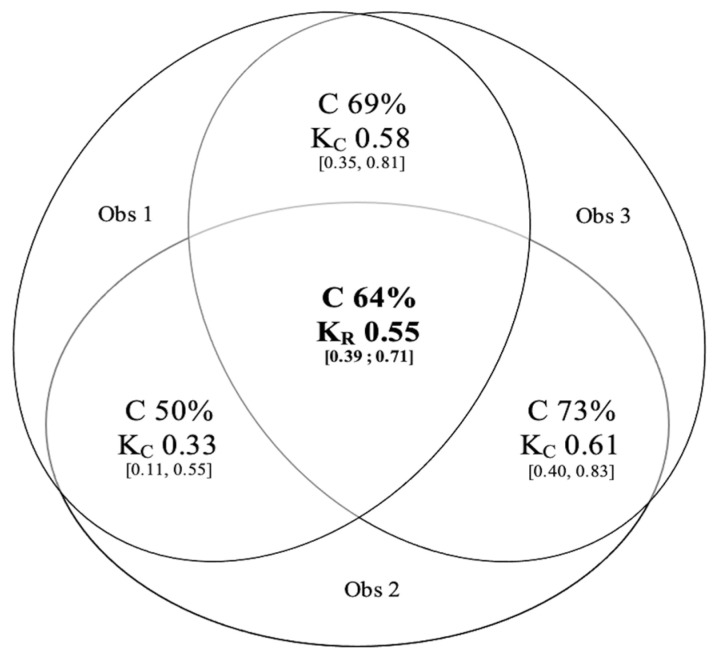
Concordance in the evaluation of the perception profile of the 26 patients between the three observers before the Delphi approach. Abbreviations: Obs, observer; C, concordance (%); K_C_, Cohen’s kappa coefficient [IC 95%]; K_R_, Randolph’s kappa coefficient [IC 95%].

**Figure 4 healthcare-10-01131-f004:**
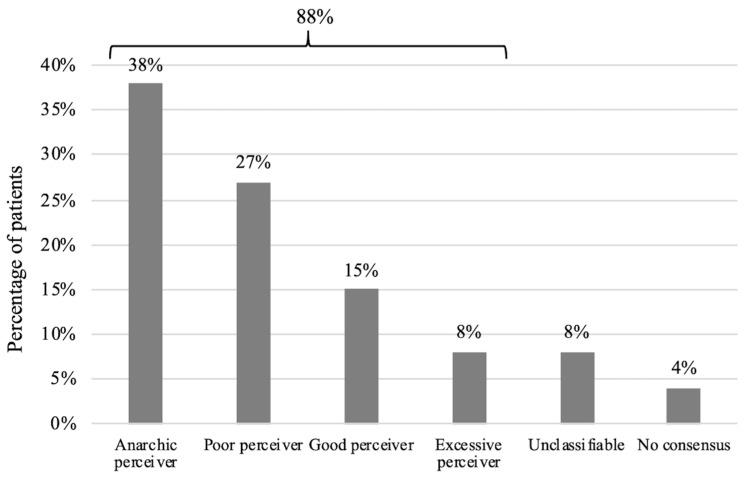
Perception profiles of the 26 asthmatic children after the Delphi approach.

**Figure 5 healthcare-10-01131-f005:**
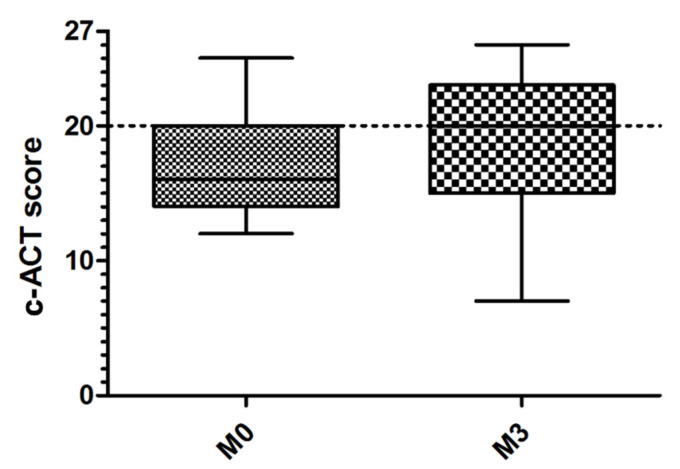
c-ACT (Childhood Asthma Control Test) score at the beginning (M0) and at the end (M3) of the study. The c-ACT score ranges from 0 (worst score) to 27 (best score). Score ≥ 20 indicates good asthma control. Boxes range from first to third quartile. The horizontal line across the box represents the median. The whiskers go from each quartile to the minimum and maximum.

**Figure 6 healthcare-10-01131-f006:**
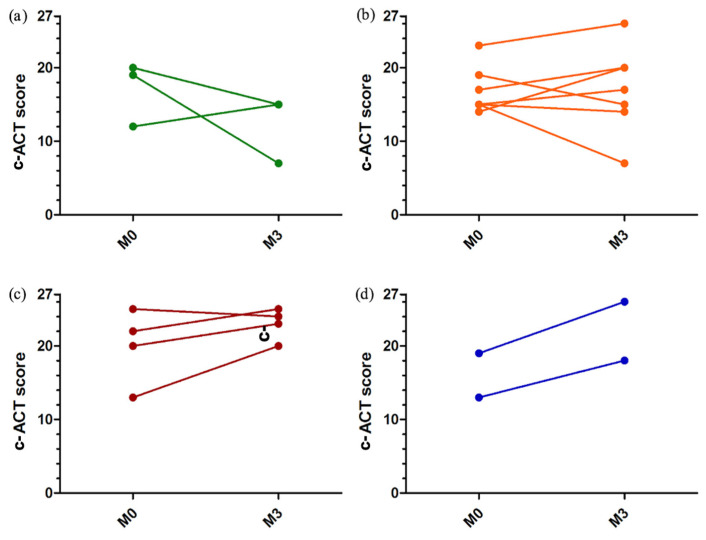
c-ACT (childhood Asthma Control Test) score at the beginning (M0) and at the end (M3) of the study according to the perception profile. Each line represents one patient. (**a**) good perceivers; (**b**) anarchic perceivers; (**c**) poor perceivers; (**d**) excessive perceivers. The c-ACT score varies between 0 (worst score) and 27 (best score). Score ≥ 20 indicates good asthma control.

**Figure 7 healthcare-10-01131-f007:**
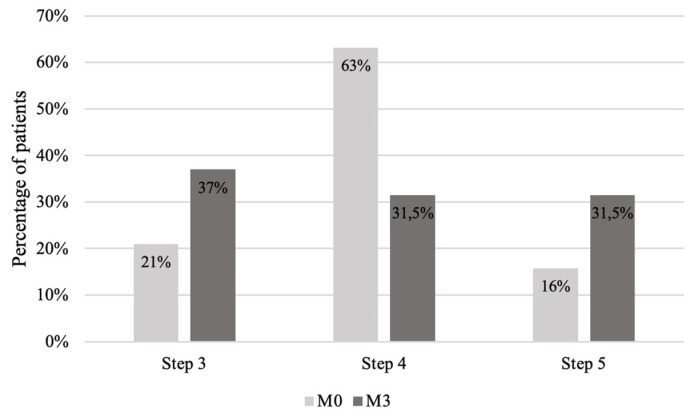
Treatment steps according to GINA (Global Initiative for Asthma) at the beginning (M0) and at the end (M3) of the study.

**Figure 8 healthcare-10-01131-f008:**
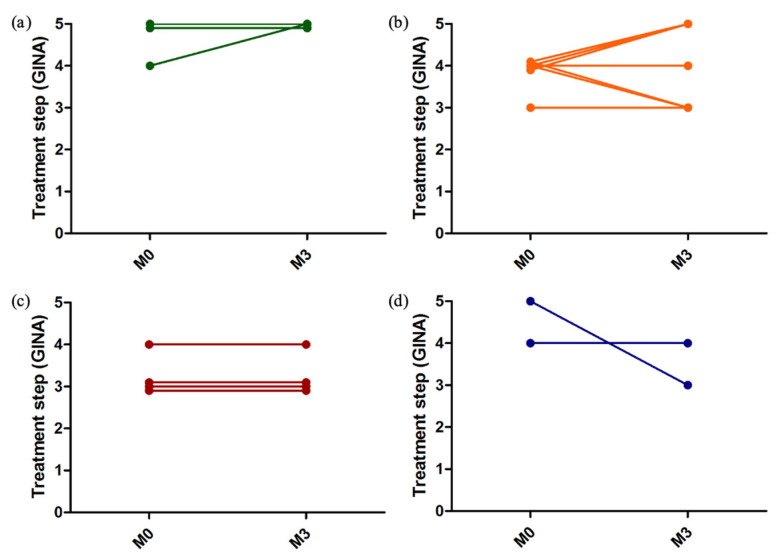
Treatment steps according to GINA (Global Initiative for Asthma) at the beginning (M0) and at the end (M3) of the study according to the perception profile. Each line represents one patient. (**a**) good perceivers; (**b**) anarchic perceivers; (**c**) poor perceivers; (**d**) excessive perceivers.

**Figure 9 healthcare-10-01131-f009:**
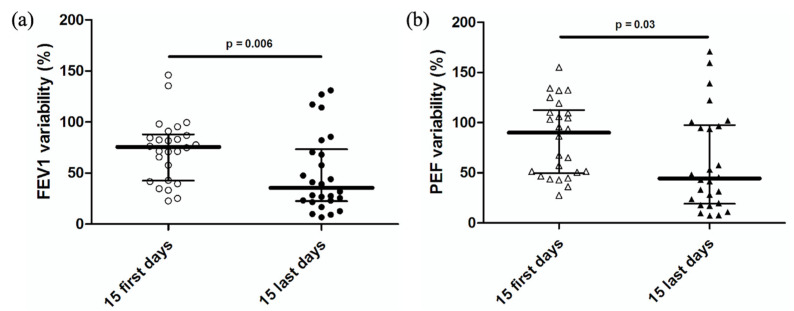
(**a**) FEV1 variability in the first 15 days and last 15 days of the 3-month follow-up. The vertical line ranges from the first quartile to the third quartile. The horizontal line in the middle represents the median. Each dot represents one patient. (**b**) PEF variability in the first 15 days and last 15 days of the 3-month follow-up. The vertical line ranges from the first quartile to the third quartile. The horizontal line in the middle represents the median. Each triangle represents one patient.

**Figure 10 healthcare-10-01131-f010:**
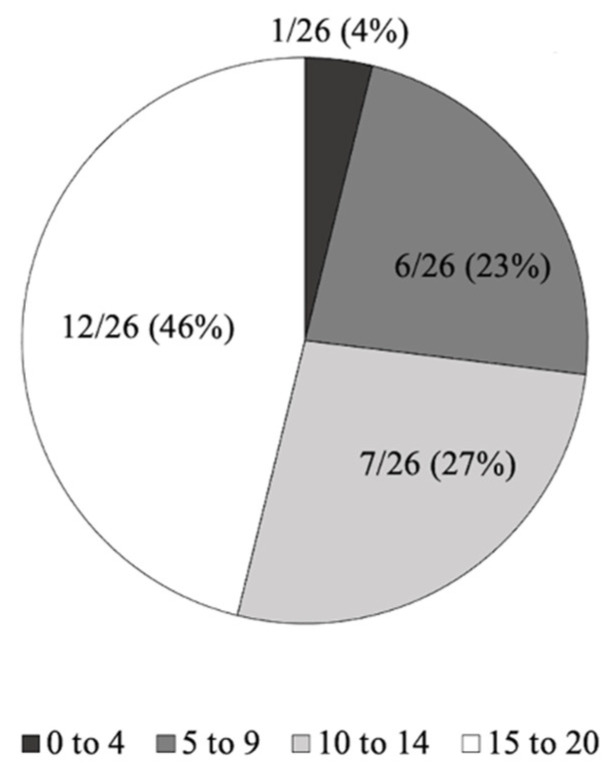
Number of records during the 10 first days of monitoring. Data in the pie charts represent the number of patients as n/N (%).

**Figure 11 healthcare-10-01131-f011:**
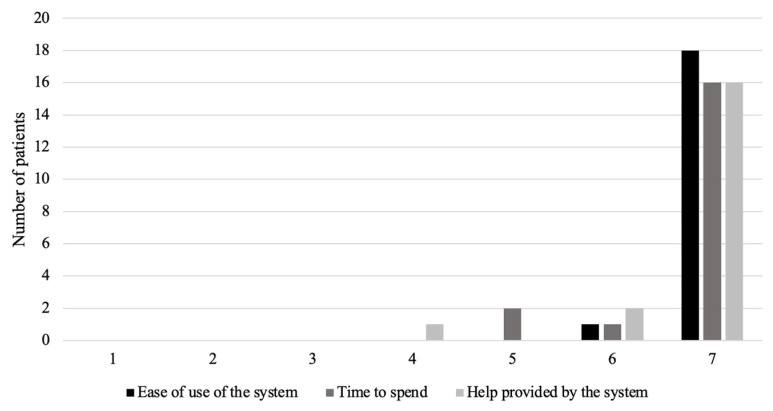
Satisfaction with Spirobank Smart^®^ follow-up according to the ASQ (After Scenario Questionnaire) after 3 months of use. Each items is rated from 1 (not satisfied) to 7 (very satisfied).

**Table 1 healthcare-10-01131-t001:** Demographic Characteristics of the Study Population.

Variable		Result
Sex	Female	13/26 (50)
Age (years)		9.5 [8.0; 11.5] (min 6–max 18)
Overweight/obesity (IOTF criteria > 25)		6/26 (23)
Age at 1st wheezing episode (years)		1.4 [0.5; 3.6]
**Environment**		
Smoking during pregnancy		5/26 (19)
Post-natal passive smoking		10/26 (38)
Living area	Rural	7/26 (27)
	Semi-rural	9/26 (35)
	Urban	10/26 (38)
Mold, dust at home		6/26 (23)
Pets	Cat	10/26 (38)
	Dog	12/26 (46)
	Rabbit	3/26 (12)
	Chicken	2/26 (8)
	Hamster	1/26 (4)
**Co-morbidities**		
Gastro-esophagal reflux disease (GERD)		1/26 (4)
Ear-nose-throat involvement (polyps, nasal obstruction, otitis media)		4/26 (15)
Obstructive Sleep Apnea Syndrome		3/26 (12)
**Atopy-allergy**		
Family History		24/26 (92)
Prick-test or serum IgE (positive for pneum-allergens)	House Dust Mite	14/26 (48)
	Pollen	10/26 (38)
	Mold	2/26 (8)
	Dog dander	5/26 (19)
	Cat dander	4/26 (15)
Atopic dermatitis		11/26 (42)
Allergic rhino-conjunctivitis		16/26 (62)
Food Allergy		8/26 (31)
**Treatment**		
SABA (as required)		26/26 (100)
ICS only		2/26 (8)
LTRA		13/26 (50)
Combined ICS + LABA		24/26 (92)
Omalizumab		3/26 (12)
Theophylline		1/26 (4)
**Severity (GINA level)**		
Moderate persistent, Level 3		4/26 (15.4)
Severe persistent, Level 4		17/26 (65.4)
Severe persistent, Level 5		5/26 (19.2)
**Asthma Control**		
≥1 medical consultation for asthma exacerbation in the previous year		19/23 (83)
≥1 hospitalization for asthma in the previous year		12/24 (50)
≥1 hospitalization in intensive care for asthma, ever		2/25 (8)
Control according to GINA score	Well controlled	2/25 (8)
	Partly controlled	11/25 (44)
	Uncontrolled	12/25 (48)
c-ACT score < 20 (uncontrolled)		15/25 (60)
Baseline FEV1 < 80% of predicted value		4/26 (15)

Note: data are presented as n/N (%), median (IQR1; IQR3). Abbreviations: c-ACT, childhood Asthma Control Test; ICS, inhaled corticosteroids; FEV1, Forced Expiratory Volume in one second; IOTF, International Obesity Taskforce; LABA, long-acting beta-agonist; LTRA, leukotriene receptor antagonist SABA, short-acting beta-agonist.

**Table 2 healthcare-10-01131-t002:** Characterization of the perception profile according to the 3 observers and after Delphi approach.

Profile	Obs 1	Obs 2	Obs 3	Results for All Patients after the Delphi Approach	Final Results for Classifiable Profiles after the Delphi Approach
Anarchic perceiver	11/26 ^§^	4/26 ^¤^	8/26 ^¤§^	10/26 (38)	10/23 (44)
Poor perceiver	5/26 *^§^	13/26 *^¤^	10/26 ^§¤^	7/26 (27)	7/23 (30)
Good perceiver	7/26 *^§^	9/26 *^¤^	5/26 ^§¤^	4/26 (15)	4/23 (17)
Excessive perceiver	2/26 ^NA^	0/26 ^NA^	2/26 ^NA^	2/26 (8)	2/23 (9)
Unclassifiable	1/26 ^NA^	0/26 ^NA^	1/26 ^NA^	2/26 (8)	-
No consensus	-	-	-	1/26 (4)	-

Note: data are presented as n/N (%). Abbreviation: Obs, observer. * *p* < 0.05 (Obs1 vs. Obs2); ^§^ *p* < 0.05 (Obs1 vs. Obs3); * *p* < 0.05 (Obs2 vs. Obs3); ^NA^ Not analyzed.

## Data Availability

The data presented in this study are available on request from the corresponding author.

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
