# Peer review of "Telehome Monitoring of Symptoms and Lung Function in Children with Asthma"

_healthcare, 2022, doi:10.3390/healthcare10061131_

Round 1

Reviewer 1 Report

  1. The perception of the majority of patients who are "anarchic" and "poor" perceivers is significant different among the experienced and the less experienced observer (11//26 compared to 4/26 and 5/26 compared to 13/26). Authors has to explain better how the parameter of the significant difference of the experience among the three observers (obs 1 35 years, obs 3 1 year) is taken into account by their statistical analysis.
  2. ACT score did not improved in either "anarchic" and "poor" perceivers who are the majority of patients. The conclusion that: "home telemonitoring ... can be used ... to help obtain better control" is not supported by the results.

Reviewer 2 Report

please find attached file

Author Response

We have adressed all the remarks and queries proposed by reviewer 2 in the new version submitted.

In addition, page 10, para, 1, the following has been added “These 16 consecutive patients were those in whom the tele-monitoring data and curves were reliably available, but not all the data corresponding to the secondary objectives.”

Reviewer 3 Report

Overall, the authors' manuscript is well-written and appropriately written. However, corrections are necessary for better communication of some criteria established by the authors.

- In section 2.2, the authors must clearly report the number of study participants, since at the end of this section the authors mention exclusion criteria, but do not specify whether any patient was excluded from the analysis.

- In section 2.2 the authors speak of /'children and adolescents/ however in section 2.3 they speak only of children.
I suggest that in section 2.2 an age group suitable for characterizing people in children and adolescents be added. And in section 2.3 the authors must say whether adolescents were included or excluded from the study.

- As mentioned throughout the manuscript, the authors reported that some methods have limitations for diagnosis. In this way, I request that the authors be able to create a topic of conclusions for the manuscript.
In this topic, the authors must report the main scientific findings of this month and must include the limitations of the method used.

- In the conclusions, the authors should also be able to report the advantages of the method used in relation to those that are available for realization.

Reviewer 4 Report

Authors have reported a challenging research topic in this manuscript. the scope can be improved on following grounds as per my observations

  1. Introduction, add Telemedicine survey and ICT roles in telemedicine for developing countries
  2. The manuscript has jumped to results directly, it is prefered to have a brief representation of proposed methodology or a block diagram
  3. Add the following in literature survey.

Ahmed, S. Syed Thouheed, K. Thanuja, Nirmala S. Guptha, and Sai Narasimha. "Telemedicine approach for remote patient monitoring system using smart phones with an economical hardware kit." In 2016 international conference on computing technologies and intelligent data engineering (ICCTIDE'16), pp. 1-4. IEEE, 2016.

Nittari, Giulio, Ravjyot Khuman, Simone Baldoni, Graziano Pallotta, Gopi Battineni, Ascanio Sirignano, Francesco Amenta, and Giovanna Ricci. "Telemedicine practice: review of the current ethical and legal challenges." Telemedicine and e-Health 26, no. 12 (2020): 1427-1437.

Ahmed, Syed Thouheed, Maheshwari Sandhya, and Sharmila Sankar. "A dynamic MooM dataset processing under TelMED protocol design for QoS improvisation of telemedicine environment." Journal of medical systems 43, no. 8 (2019): 1-12.

Loeb, Alexander E., Sandesh S. Rao, James R. Ficke, Carol D. Morris, Lee H. Riley III, and Adam S. Levin. "Departmental experience and lessons learned with accelerated introduction of telemedicine during the COVID-19 crisis." The Journal of the American Academy of Orthopaedic Surgeons (2020).

Round 2

Reviewer 3 Report

After the modifications the ms can be accepted for publication.